# Transcriptional repression by FACT is linked to regulation of chromatin accessibility at the promoter of ES cells

Constantine Mylonas[1], Peter Tessarz[1,2]

The conserved and essential histone chaperone, facilitates chromatin transcription (FACT), reorganizes nucleosomes during DNA transcription, replication, and repair and ensures both efficient elongation of RNA Pol II and nucleosome integrity. In mammalian cells, FACT is a heterodimer, consisting of SSRP1 and SUPT16. Here, we show that in contrast to yeast, FACT accumulates at the transcription start site of genes reminiscent of RNA polymerase II profile. Depletion of FACT in mouse embryonic stem cells leads to deregulation of developmental and pro-proliferative genes concomitant with hyper-proliferation of mES cells. Using MNase-seq, Assay for Transposase-Accessible Chromatin sequencing, and nascent elongating transcript sequencing, we show that up-regulation of genes coincides with loss of nucleosomes upstream of the transcription start site and concomitant increase in antisense transcription, indicating that FACT impacts the promoter architecture to regulate the expression of these genes. Finally, we demonstrate a role for FACT in cell fate determination and show that FACT depletion primes embryonic stem cells for the neuronal lineage.

## Introduction

The basic functional unit of chromatin is the nucleosome consisting of around 147 bp of DNA wrapped around an octamer of histone proteins—two copies each of histones H2A, H2B, H3, and H4. In vitro, chromatinized DNA templates are refractory to transcription, suggesting that the nucleosome might provide a barrier for the elongating RNA polymerase. Using elegant biochemical fractionation assays coupled to in vitro transcription assays, facilitates chromatin transcription (FACT) was initially characterised as a factor that alleviated the repressive nature of chromatin in vitro (Orphanides et al, 1999). Meanwhile, it has been demonstrated that FACT can cooperate with all RNA polymerases in the cell and ensure both efficient transcription elongation and nucleosome integrity. Both FACT subunits are highly conserved across all eukaryotes with the exception of an HMG-like domain present in SSRP1 but absent in

the yeast homolog Pob3. In yeast, an HMG domain protein named Nhp6 has been proposed to provide the DNA binding capacity of FACT (Formosa et al, 2001).

The molecular basis for FACT activity has long remained elusive. However, recent biochemical and structural studies are starting to elucidate how FACT engages nucleosomes (Winkler & Luger, 2011; Hondele et al, 2013; Hsieh et al, 2013; Kemble et al, 2015). Via its several domains, FACT binds to multiple surfaces on the nucleosome octamer and acts by shielding histone–DNA interactions. Initially, it was proposed that FACT would evict an H2A/B dimer from the nucleosome in front of the polymerase and then reinstate nucleosome integrity in its wake. However, other data suggest that this dimer replacement is not part of FACT function as it leaves the histone composition of the nucleosome intact (Formosa, 2012). Based on recent biochemical data (Hsieh et al, 2013), a model emerges in which RNA Pol II enters the nucleosome and partially uncoils the nucleosomal DNA. At the same time, FACT binds to the proximal and distal H2A/H2B dimer, and these FACT–dimer interactions facilitate nucleosome survival.

Although the genetics and biochemistry of FACT are relatively well understood, it is not known whether cell-type dedicated functions are conferred by this histone chaperone. Interestingly, genome-wide expression analyses across cell and tissue types implicate a role of FACT in maintaining an undifferentiated state. Depletion of FACT subunits leads to growth reduction in transformed but not in immortalized cells (Garcia et al, 2013), indicating that FACT is essential for tumour growth but not for proliferation of untransformed cells. Finally, FACT regulates the expression of Wnt target genes during osteoblast differentiation in mesenchymal stem cells and its deletion leads to a differentiation skew (Hossan et al, 2016). Taken together, these data suggested a more specific role for the FACT complex in undifferentiated cells as previously assumed.

Recent studies have demonstrated that RNA Pol II can transcribe in both sense and antisense directions near many mRNA genes (Kwak et al, 2013; Mayer et al, 2015). At these so-called bidirectional promoters, RNA Pol II initiates transcription and undergoes promoter-proximal pausing in both the sense (at the protein-coding transcription start site [TSS]) and antisense orientation (Kwak et al, 2013; Mayer et al, 2015). Divergent transcription is often

[1]Max Planck Institute for Biology of Ageing, Cologne, Germany    [2]Cologne Excellence Cluster on Cellular Stress Responses in Ageing Associated Diseases, University of Cologne, Cologne, Germany

Correspondence: ptessarz@age.mpg.de

found at mammalian promoters that are rich in CpG content, but lack key core promoter elements such as the TATA motif (Scruggs et al, 2015). A broader nucleosome free region (NFR) in the promoter region is often accompanied by divergent transcription, and can lead to binding of more transcription factors (TFs) resulting in higher gene activity (Scruggs et al, 2015).

Here, we have confirmed an indispensable role of FACT in un-differentiated cells based on the expression levels of both FACT subunits and, thus, chose mouse embryonic stem cells as a model to investigate how FACT might shape the transcriptome and maintain an undifferentiated state. To achieve this, we performed chromatin immunoprecipitation and sequencing (ChIP-seq) and RNA-seq to identify genes bound and regulated by FACT. To address at a mechanistic level how FACT might regulate transcription in embryonic stem (ES) cells, we combined this analysis with MNase digestion of chromatin coupled to deep sequencing (MNase-seq), assay for transposase-accessible chromatin using sequencing (ATAC-seq), and nascent elongating transcript sequencing (NET-seq). Using these approaches, we have identified a specific gene cluster comprising genes involved in embryogenesis/neuronal development that are up-regulated upon FACT depletion. In addition, we observed a concomitant increase in chromatin accessibility around the TSS, suggesting that maintenance of nucleosomes at this position by FACT is part of the mechanism how FACT impacts on the regulation of these genes. Finally, our data support a role of FACT in the maintenance of a pluripotent state by showing that its depletion leads to faster differentiation into the neuronal lineage.

# Results

## Occupancy of FACT correlates with marks of active gene expression

High expression of FACT has been associated with stem or less-differentiated cells (Garcia et al, 2011). Indeed, we were able to confirm that low FACT levels correlate with highly differentiated cell lines as opposed to stem and cancer cells (Fig S1A). In addition, differentiation of murine ES cells into terminally differentiated cardiomyocytes (Wamstad et al, 2012) reveals that FACT levels diminish throughout the course of differentiation (Fig S1B). Thus, we chose to explore how FACT contributes to the transcriptome of undifferentiated cells using mouse ES cells. Initially, we applied to mESCs a ChIP-seq assay to identify potential DNA-binding regions for both FACT subunits. Subsequently, we examined FACT co-enrichment with several other TFs, histone marks, and chromatin remodellers over the gene body area of all uniquely annotated protein-coding genes (n = 11,305). High correlation scores were observed between SSRP1, SUPT16, H3K4me3, H3K27ac, and Pol II variants (Pol II S5ph and Pol II S2ph), confirming the role of FACT in active gene expression (Figs 1A and S1E). A good correlation was also observed between both FACT subunits and Chd1, in line with data demonstrating physical interaction and co-localization in mammalian cells (Kelley et al, 1999). However, only a moderate correlation was observed between FACT and H3K36me3 on a genome-wide level despite the fact that H3K36me3 directly recruits FACT to

actively transcribed genes (Carvalho et al, 2013). We suspect that the strong enrichment of FACT subunits around the TSS might mask this potential correlation. Nevertheless, FACT subunits also co-localize to the gene body of actively transcribed genes and enrich towards the transcription end site, similarly to H3K36me3 (Fig S1C and D). Pearson's correlation among FACT and active marks remained elevated when we focused on promoter and enhancer regions (n = 19,461) (Fig 1B). Both subunits displayed very similar binding pattern to each other over the TSS of all the annotated genes and were tightly linked to H3K4me3 levels (Fig 1C).

## Regulation of gene expression by FACT

To investigate how FACT orchestrates transcriptional regulation in mESCs, we depleted SSRP1 levels using short hairpin RNAs (shRNA; Fig S2A). Importantly, this also led to a simultaneous depletion of SUPT16 levels as assessed by mass spectrometry (Table S6). This interdependence of the two FACT subunits has been observed before (Garcia et al, 2013). Surprisingly, we observed an increase in mESC proliferation following Ssrp1 knockdown (KD) as measured by proliferation rate via metabolic activity measurement (MTT) cell proliferation assays using independent shRNAs (Figs 2A and S2B). This is in contrast to previously published data from tumour cell lines, in which proliferation rates decrease, and also from terminally differentiated cells, where FACT depletion has no effect on proliferation (Garcia et al, 2013). Subsequently, we sequenced the whole transcriptome (RNA-seq). In total, we characterized 3,003 differentially expressed genes: 1,655 down-regulated and 1,348 up-regulated (Fig 2B). Down-regulated genes were overrepresented for pathways involved in development, whereas up-regulated genes were involved in metabolic processes and positive regulation of proliferation (Fig 2C), indicating that the change in the transcriptome accounts for the faster proliferation rates. These results suggest that FACT impacts developmental processes and negatively controls cell proliferation in mES cells by controlling gene expression patterns. A low correlation (Pearson's R = 0.11) was observed between the coverage of SSRP1 (ChIP-seq) and the gene fold change (RNA-seq) of those genes in the Ssrp1 KD (Fig 2D), indicating that FACT binding alone is not a predictor for gene expression changes. Taking these findings together, FACT can work directly as an enhancer or repressor of transcription in mES cells.

Given the high correlation of FACT with H3K4me3 (Fig 1A) and to understand how the transcriptional changes might be linked to differences in recruitment of transcriptional regulators, we performed an IP for H3K4me3 followed by mass spectrometry both in control and SSRP1-depleted ES cells (Fig S3A and Table S5). We observed an increased binding of Oct4 and Sox2 to H3K4me3 in the Ssrp1 KD state, in line with the observation that FACT depletion impacts developmental processes. Interestingly, we observed reduced binding of many splicing factors on H3K4me3 in the absence of FACT (Fig S3A). Differential splicing analysis between control and Ssrp1 KD conditions confirmed in total 356 exon skipping/inclusion and 97 intronic retention events following FACT depletion, of which around 50% are direct targets of SSRP1 (Fig S3B and C). However, at present, it is not clear whether the effects on splicing factor binding and splicing pattern are directly and mechanistically coupled to FACT depletion. Interestingly, a fraction of the differential gene isoforms generated in the Ssrp1 KD group is overrepresented in limbic system

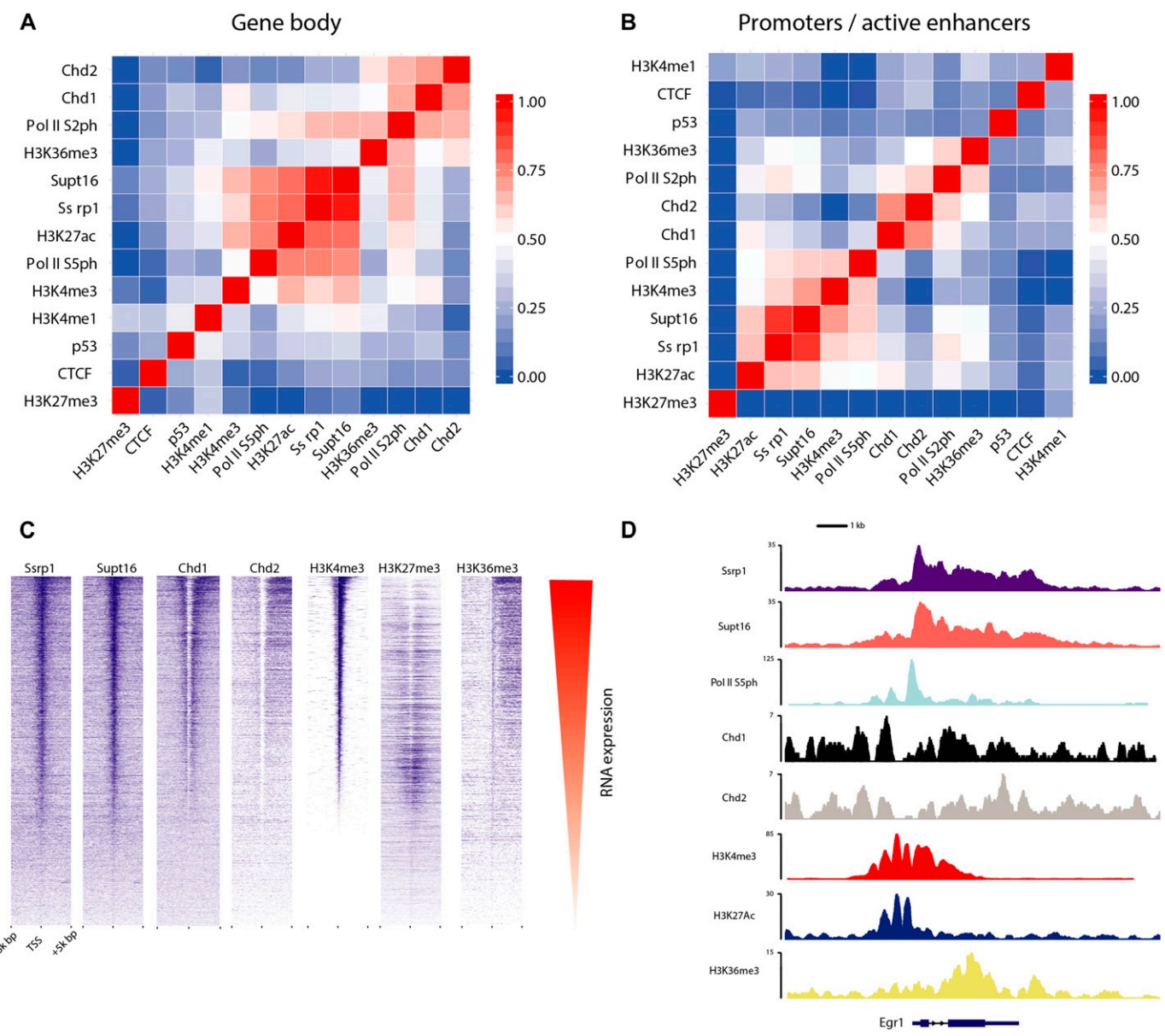

**Figure 1. Correlated occupancies across FACT-bound regions.**
**(A)** Heatmap representing Pearson's correlation between FACT subunits (SSRP1 and SUPT16) and other factors over the gene body area of all uniquely annotated protein-coding genes (*n* = 11,305). **(B)** Same as (A) but for promoter/active enhancer regions (*n* = 19,461) characterized by high H3K27ac and/or Pol II density. **(C)** Distribution of FACT and other factors (ChIP-seq tags indicated in blue) over the TSS of 11,305 unique RefSeq genes, sorted by H3K4me3 levels. Coinciding RNA expression levels are shown in red. **(D)** Distribution of several ChIP-seq datasets over a single gene (*Egr1*).

and dendrite development pathways (Fig S3D), suggesting that genes involved in neuronal development might be influenced by FACT.

### Depletion of FACT induces very specific changes in chromatin accessibility

Because FACT is responsible for the remodelling of nucleosomes in front of RNA polymerase and the reestablishment of nucleosome integrity in its wake (Formosa, 2012), we speculated whether some of the observed transcriptional alterations could be connected to changes in nucleosome occupancy upon depletion of FACT.

Mononucleosome-sized DNA fragments upon treatment with MNase (135–170 bp) were purified from control and *Ssrp1*-depleted conditions and sequenced (Fig S4A and B). Nucleosome occupancy was plotted for four different gene classes according to the presence of SSRP1 in the control group (ChIP-Seq) and their relative gene fold change (RNA-seq) in the *Ssrp1* KD state. Overall, we observed little changes in nucleosome occupancy genome wide (Fig 3A). Genes that are down-regulated in the *Ssrp1* KD ("down-regulated" class) and bound by FACT exhibit a global mononucleosomal shift by a few nucleotides right after the +1 nucleosome. Up-regulated genes showed a loss of nucleosome occupancy in the gene body area regardless of FACT-bound

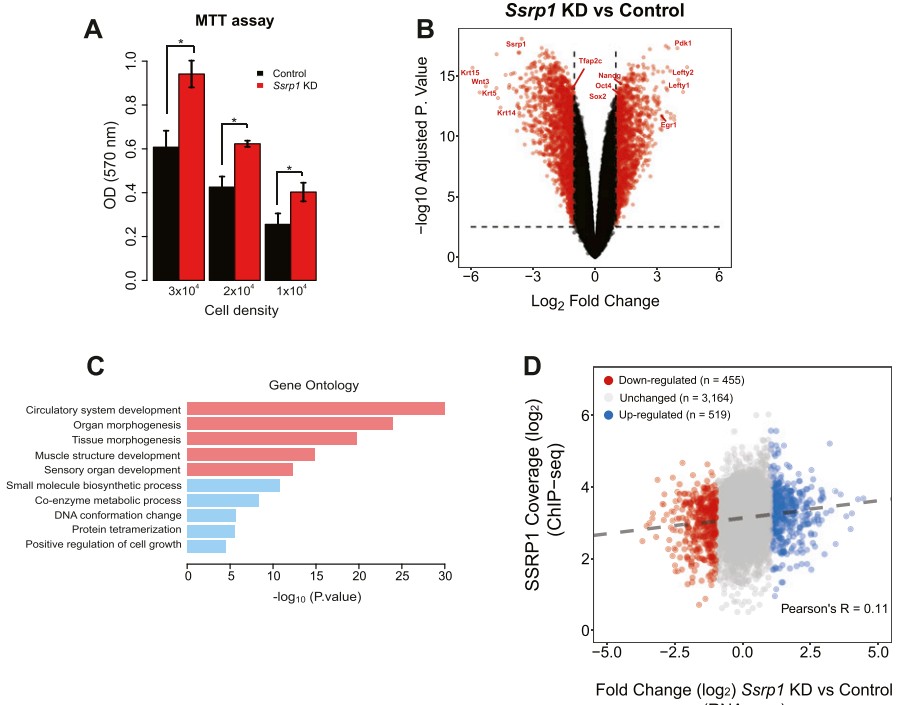

**Figure 2. Regulation of gene expression by FACT.**
**(A)** MTT assay assessing cell metabolic activity in mESCs at different cell densities following depletion of FACT levels. Values are mean and SE of three independent transfection experiments are displayed. Significance was calculated via a two-tailed *t* test (*P* < 0.05). **(B)** Volcano plot of differentially expressed genes between the control and KD group. Values with logFC > 1 or logFC < −1 and adjusted *P*-value < 0.01 are highlighted in red. **(C)** Gene ontology analysis of all differentially expressed genes (red: pathways for down-regulated genes and blue: pathways for up-regulated genes). **(D)** Scatterplot of log (SSRP1 coverage) (ChIP-seq) over logFC (RNA-seq). Numbers for up-, down-, and non-changing genes are given. Correlation between SSRP1 coverage and gene expression change (fold change) is indicated by the dashed line.

status (non-SSRP1 and SSRP1 targets) (Fig 3A and B). However, specifically in up-regulated genes bound by FACT ("up-regulated" class), we observed a significant loss of nucleosomes upstream of the TSS (Fig 3A and B). This difference in nucleosome occupancy at the promoter region is highly reproducible among the different replicates (Fig S4C). Splitting the up-regulated genes by the amount of H3K4me3 levels (k-means clustering) as a proxy for gene expression levels also revealed that the loss of nucleosomes at the promoter is more profound over the promoters of lowly expressed/repressed genes (control state) (Fig S4C). The observed nucleosome-depleted regions were different between up- and down-regulated genes. Such architectural differences have been previously attributed to different levels of GC frequency. Indeed, GC frequency over SSRP1 targets was higher and broader in the "down-regulated" class corroborating a more open chromatin state (Fenouil et al, 2012) (Figs 3A and S4D).

To confirm this difference in chromatin accessibility using an additional approach, we performed ATAC-seq in control and *Ssrp1*-depleted ES cells (Fig 3C). In line with the observations of the MNase-seq experiments, we observed a statistically significant increase (*P* < $10^{-10}$) in chromatin accessibility in the absence of FACT upstream of the promoter region of FACT-bound, up-regulated genes (Fig 3C–E). In combination with the RNA-seq data, this reduction in nucleosome occupancy (and subsequently increase in chromatin accessibility) at the TSS suggests that FACT might act as a repressor by enabling a more closed chromatin conformation state at promoter regions.

### Gain in chromatin accessibility upon FACT depletion upstream of the TSS correlates with an increase in antisense transcription

Over the last decade, it has become apparent that promoters can drive expression of sense and antisense RNAs, with proximally

paused RNA Pol II on both strands (Seila et al, 2008; Jonkers et al, 2014). In vitro, FACT has been demonstrated to facilitate transcription through chromatinized templates (Orphanides et al, 1999) and reduces pausing of the elongating polymerase when it encounters nucleosomes (Hsieh et al, 2013). In yeast, depletion of Spt16 leads to up-regulation of antisense transcription from gene-internal cryptic promoters (Feng et al, 2016). Thus, to understand how the observed changes in chromatin accessibility would impact transcription initiation and to get more mechanistic insight into how FACT might dampen expression of genes in mES cells, we performed NET-seq (Mayer & Churchman, 2016) (Fig S5A), a method that allows quantitative, strand-specific, and nucleotide resolution mapping of RNA Pol II.

Initially, we sought to determine whether nascent transcription positively correlates with mRNA levels. A higher correlation of nascent RNA–mRNA expression and a significantly higher slope (*P* < $10^{-5}$) was observed over SSRP1-target regions in the control state, suggesting higher levels of Pol II pausing and mRNA levels in the presence of FACT (Fig 4A). Nevertheless, in the *Ssrp1* KD state, the SSRP1-bound regions maintained a higher slope, suggesting that pausing and elongation speed of RNA Pol II are not controlled entirely by FACT alone (Fig S5B). To confirm this, we measured the travelling ratio of RNA Pol II over down-regulated and up-regulated genes. Indeed, "up-regulated" SSRP1-bound genes show a lower travelling ratio overall. Interestingly, under our experimental conditions, we did not observe a significant difference among this group of genes following FACT depletion (control to *Ssrp1* KD comparison; Fig 4B).

Next, we assessed RNA Pol II pausing and directionality over up-regulated genes. NET-seq density plots identified that FACT targets displayed higher levels of promoter-proximal RNA Pol II than

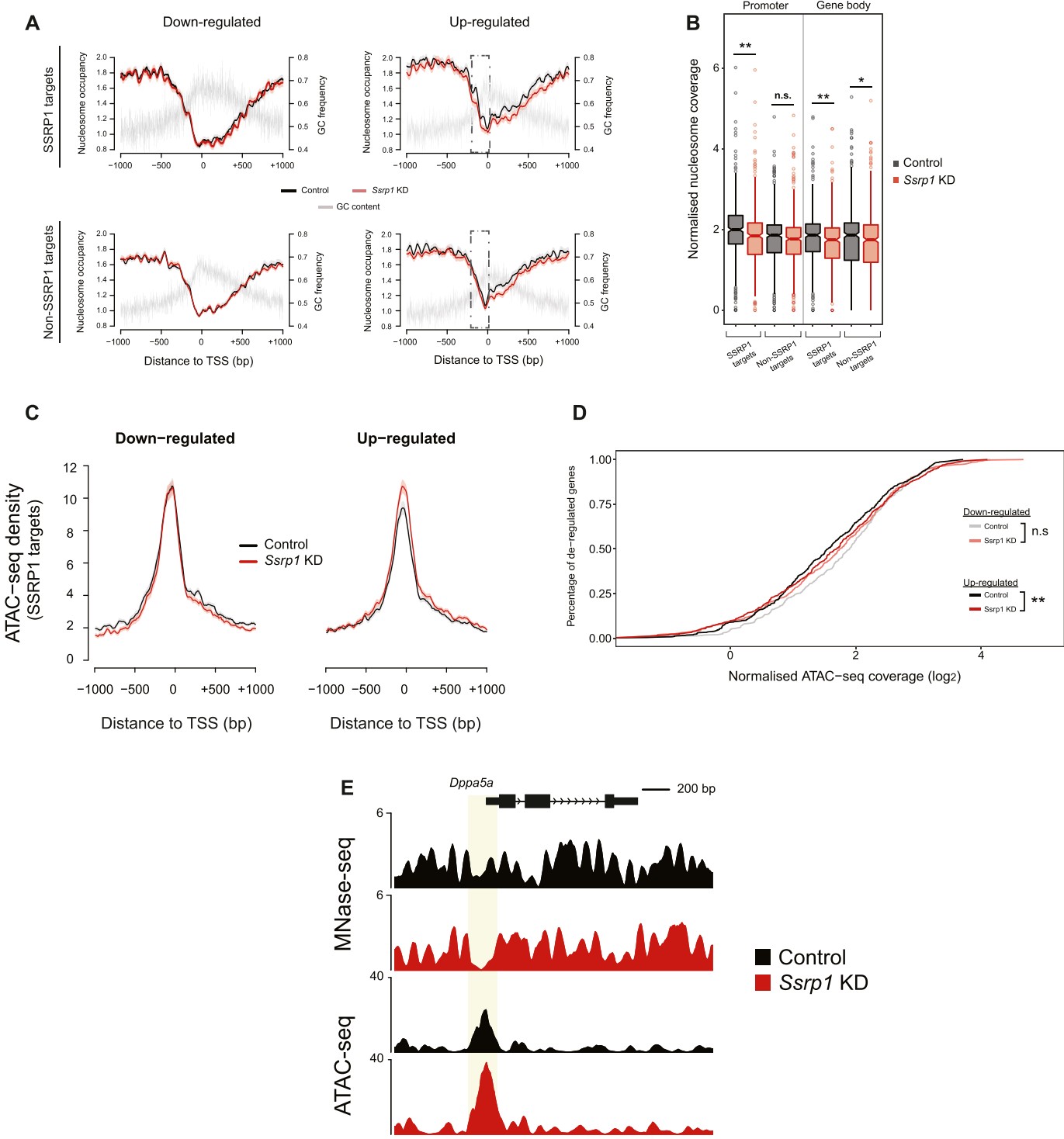

**Figure 3. Regulation of gene expression by FACT through chromatin accessibility.**
**(A)** Nucleosome occupancy of all deregulated genes. Datasets are split by their FACT occupancy status (SSRP1 and non-SSRP1 targets) and their relative transcriptional direction ("down-regulated" and "up-regulated") following SSRP1 depletion. Solid lines indicate the mean values, whereas the shading represents the SE of the mean. **(B)** Boxplots measuring the nucleosome occupancy ($\log_2$) over promoters and gene body area of up-regulated genes (**$P < 0.001$, *$P < 0.05$, and n.s., not significant). The assessed promoter region is shown in dashed boxes indicated in (A). Significance was calculated using the Welch two-sample $t$ test. **(C)** Metaplot of open chromatin assessed by ATAC-seq among down-regulated and up-regulated genes both in control and *Ssrp1* KD conditions. **(D)** Cumulative distribution of ATAC-seq density for genes and conditions displayed in (C). Significance was calculated using the Welch two-sample $t$ test (**$P < 10^{-9}$ and n.s., not significant). **(E)** Interrogation of nucleosome occupancy (MNase-seq) and chromatin accessibility (ATAC-seq) over the *Dppa5a* gene promoter for control and *Ssrp1* KD conditions. Changes in nucleosome occupancy and chromatin accessibility are highlighted in yellow.

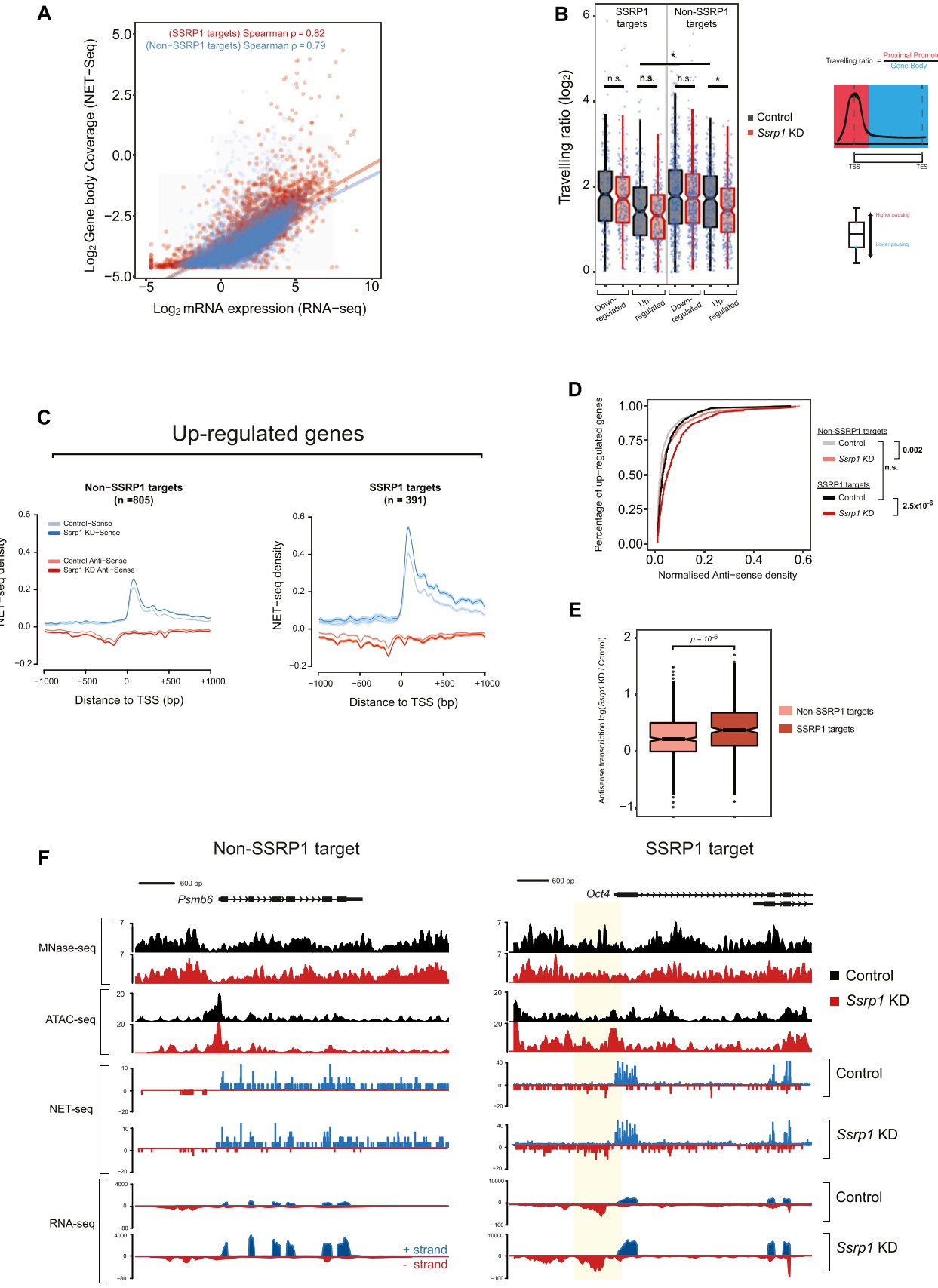

SSRP1-unbound promoters (Fig 4C and D). Upon KD of FACT, SSRP1 targets displayed an increase ($P < 10^{-6}$) in divergent transcription compared with the non-SSRP1 targets (Fig 4E). This occurred precisely at locations where nucleosomes were depleted upon KD of FACT (Fig 4F). No change in antisense transcription was observed for down-regulated (Fig S6A and B) or unchanged (Fig S6C and D) genes, suggesting that the presence of FACT over a specific gene class (up-regulated genes) decreases the rate of antisense transcription by maintaining higher nucleosome density upstream of the TSS.

A correlation between loss of nucleosomes upstream of the TSS and increase in antisense and sense transcription has recently been reported to occur in mammalian cells (Scruggs et al, 2015). Furthermore, this study showed that antisense transcription can lead to a more open chromatin structure, enabling increased binding of TFs, which is favourable for sense transcription.

**ES cells differentiate more efficiently into the neuronal lineage upon FACT depletion**

Finally, we wanted to investigate whether the transcriptional changes induced by depletion of FACT have physiological consequences. We tested this by differentiating mES cells into the neuronal lineage. The rationale for this approach stems from previous studies that pinpoint a specific role for FACT in neurons (Neumüller et al, 2011; Vied et al, 2014) and from the gene ontology enrichment for neuronal terms that we obtained from mRNA isoform analysis (Fig S3D). We induced differentiation of ES cells towards a neuronal lineage via embryoid body formation and treatment with retinoic acid (RA) (Bibel et al, 2007). We created early-stage neural precursor cells (NPCs; 3 d into the differentiation process) and interrogated the whole transcriptome via RNA-seq. We identified that in these early-stage NPCs, expression of key neurogenesis markers (*Pax6*, *Nes*, and *Tubb3*) increases, whereas FACT mRNA levels and key pluripotency factors are yet unchanged and still maintained at a high level (Fig 5A). A quarter of the up-regulated genes in ES cells after *Ssrp1* KD overlaps with the up-regulated genes instigated by neuronal differentiation ($P < 10^{-13}$, Fisher's exact test; Fig 5B) and are overrepresented in pathways involved in neuronal development. Similar to our previous observations, β3-tubulin (*Tubb3*) (SSRP1-bound gene), as an example for neurogenesis genes up-regulated upon FACT depletion, shows higher chromatin accessibility levels at the promoter region upon

KD of FACT. This opening of the promoter is accompanied by an increase in antisense transcription (Fig 5C).

We then depleted *Ssrp1* levels at the onset of neuronal differentiation and performed immunofluorescence for neurogenesis (β3-tubulin) and dendritic (MAP2) markers at the same time point as the RNA-seq experiment. *Ssrp1* KD caused a substantial increase in the expression of those markers as measured by immunofluorescence, indicating that loss of FACT function primes ES cells for the neuronal lineage and enhances early neuronal differentiation (Fig 5D).

# Discussion

In this study, we have addressed the role of the histone chaperone FACT in mouse ES cells. In contrast to the genomic profile identified for *Saccharomyces cerevisiae* FACT, where the protein occupancy is depleted at the TSS and accumulates in the gene body (True et al, 2016), the genomic profile of mammalian FACT over active genes is reminiscent of a profile of the Ser5 phosphorylated form of RNA Pol II. This recruitment to the TSS might reflect binding of FACT to RNA Pol II. A similar profile for SSRP1 has been reported recently in HT1080 cells (Garcia et al, 2013).

In general, FACT depletion does not lead to gross alterations of the nucleosomal landscape as measured by MNase- and ATAC-seq. In particular, genes down-regulated upon FACT depletion only show a slight shift of nucleosomes, similar to what has been observed in yeast upon FACT inactivation (Feng et al, 2016). It is tempting to speculate that the reason for down-regulation lies in the originally described function of FACT to help passage of RNA Pol II through chromatin (Orphanides et al, 1999) and its depletion makes this process less efficient. FACT-bound genes that are up-regulated upon *Ssrp1* depletion show a significant alteration in nucleosomal occupancy around the TSS. FACT depletion leads to loss of nucleosomes and increased rates of bi-directional nascent transcription, suggesting that these genes are usually dampened or repressed (in case of silent genes) by the maintenance of nucleosomes at these sites. The loss of nucleosomal occupancy upon depletion of FACT goes hand-in-hand with an increase in antisense transcription. Based on the data presented here, we cannot determine if the loss of nucleosomes precedes up-regulation of antisense transcription or vice versa. Also, it is not clear whether this is driven by FACT alone or in combination with RNA polymerase and/or chromatin remodellers. However, it is clear that this

**Figure 4. Regulation of RNA Pol II directionality by FACT.**
**(A)** Scatterplots of log gene body coverage (NET-seq) versus log mRNA expression (RNA-seq) for SSRP1 ($n$ = 4,576) and non-SSRP1 ($n$ = 8,844) target regions in the control state ($z$-score = 5.3, $P < 10^{-5}$). **(B)** Measure of Pol II pause/release. Travelling ratio is defined as NET-seq density of proximal promoter versus gene body area. The log-transformed travelling ratio for each gene class is displayed with boxplots. The Welch two-sided $t$ test was used to calculate significance between control and *Ssrp1* KD (*$P < 0.05$, n.s., not significant). **(C)** NET-seq density plots (control and *Ssrp1* KD group) of up-regulated genes split by FACT-bound status (non-SSRP1 and SSRP1 targets). Solid lines indicate mean values, whereas the shading represents the 95% confidence interval. **(D)** Cumulative distribution of antisense transcription (NET-seq) in a window 1,000 bp upstream of the TSS. The Welch two-sided $t$ test was used to calculate significance between control and *Ssrp1* KD among non-SSRP1 and SSRP1 targets. **(E)** Boxplots assessing fold change (*Ssrp1* KD versus control) in antisense transcription (NET-seq) in a window 1,000 bp upstream of the TSS. The Welch two-sided $t$ test was used to calculate significance between non-SSRP1 and SSRP1 targets. **(F)** Nucleosome occupancy (MNase-seq), open chromatin (ATAC-seq), and transcriptional activity (NET-seq/RNA-seq) over an SSRP1 (*Oct4*) and non-SSRP1 (*Psmb6*) target gene between control and *Ssrp1* KD conditions. Nucleosomal loss and increase in antisense transcription at the *Oct4* promoter is highlighted in yellow.

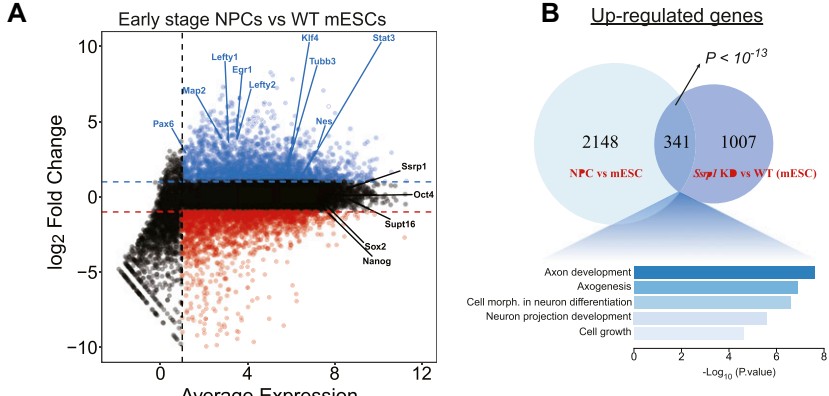

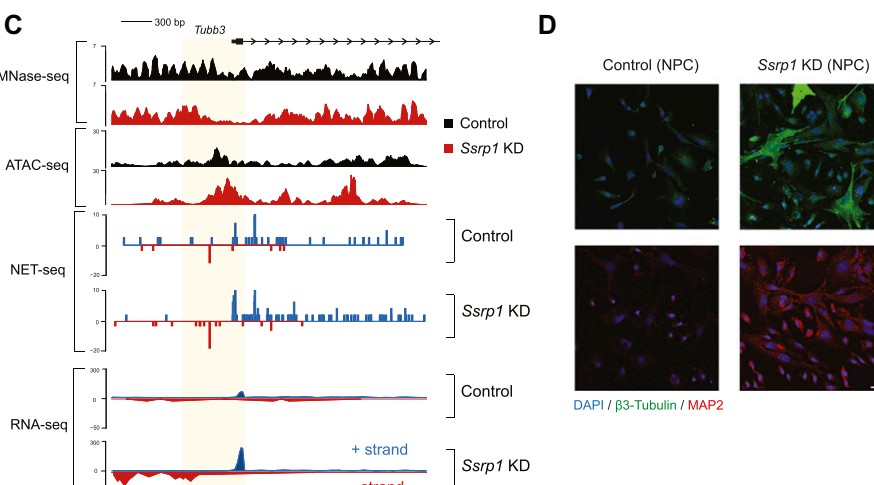

**Figure 5. FACT regulates neurogenesis through Pol II/nucleosome dynamics.**
**(A)** MA plot depicting differential expression in NPCs versus WT ES cells. Up-regulated genes are highlighted in blue, whereas down-regulated genes are highlighted in red. **(B)** Venn diagram showing the overlap of up-regulated genes between NPC versus mESCs and control versus *Ssrp1* KD mESCs. **(C)** Interrogation of nucleosome occupancy (MNase-seq), chromatin accessibility (ATAC-seq), and transcriptional activity (NET-seq/RNA-seq) over the *Tubb3* gene promoter for control and *Ssrp1* KD conditions. Changes in nucleosome occupancy, chromatin accessibility, and Pol II occupancy are highlighted in yellow. **(D)** Immunofluorescence analysis of early-stage NPCs following *Ssrp1* depletion: (blue) DAPI, nuclei; (green) β3-tubulin (Tubb3), neurons; and (red) MAP2, dendrites. Scale is 20 μm.

observed effect is very specific to SSRP1-bound genes that are up-regulated upon depletion of FACT. One should note, however, that this gene class shows low levels of antisense transcription (Fig 4C and D). Therefore, one plausible model would be that FACT is required on these promoters to reinstate nucleosomes after initiation of antisense transcription. Depletion of FACT would lead to loss of this function and loss of nucleosomes, which in turn would drive higher levels of antisense transcription. It is of interest to note that FACT depletion in *S. cerevisiae* by using a thermosensitive allele of *spt16* also leads to up-regulation of sense/antisense transcription. However, this occurs at cryptic promoters within the coding region of the gene because of a defect in reestablishing chromatin structure after passage of the elongating polymerase (Feng et al, 2016). Given the differences of FACT occupancy between mammals (this study; Garcia et al, 2013) and yeast (True et al, 2016), this might reflect evolutionary differences between mammalian and yeast FACT.

This scenario described for mammalian FACT would lead to a wider NFR and allow more efficient recruitment of TFs and RNA polymerase. In addition, the torque generated by two divergently elongating RNA Pol II molecules can create sufficient negative supercoiling density in the DNA between the two promoters, which is known to increase RNA Pol II transcription efficiency (Seila et al, 2009). Taken together, we have shown that FACT can function both as an enhancer and a repressor of transcription. The repressive function of FACT correlates well with nucleosomal occupancy at the TSS and suppression of antisense transcription.

FACT expression correlates with the differentiation state of the cell, being highest in undifferentiated and lowest in terminally differentiated cells. This cannot be simply explained by differences in proliferation rates as, e.g., NIH-3T3 also exhibits low levels of FACT expression but proliferates comparably with mouse ES cells. These observations suggest that FACT assists to maintain a chromatin/transcription state that allows self-renewal. Indeed, depletion of FACT leads to an imbalance of the ES cell transcriptome. On the one hand, pro-proliferative genes are up-regulated and lowly expressed developmental factors are further down-regulated, leading to the hyper-proliferation of ES cells. Moreover, the FACT-depleted gene signature has a large overlap with gene expression changes observed upon differentiation into the neuronal lineage. Interestingly, a comparison of expression patterns in the early developing mouse brain identified a set of only 13 genes, including *Ssrp1* with high correlation of expression in the proliferating cells of the ventricular zone of the neocortex at early stages of development (Vied et al, 2014). This is a transient embryonic layer of tissue containing neural stem cells (Rakic, 2009)

and a place for neurogenesis during development dependent on the Notch pathway (Rash et al, 2011). Similar to our study, hyper-proliferation in a stem cell compartment upon FACT depletion has been observed before. *Drosophila* neuroblasts hyper-proliferate upon deletion of SSRP1, suggesting that it is involved in the regulation of balancing neuroblast self-renewal and differen-tiation (Neumüller et al, 2011). A very recent report also high-lights the role of FACT in assisting cell fate maintenance. Using a genetic screen in *Caenorhabditis elegans*, all FACT subunits were identified as barriers for cellular reprogramming of germ cells into the neuronal lineage (Kolundzic et al, 2017 *Preprint*). Comparable with our results, the authors did not observe major chromatin architecture alterations but observed larger colonies during reprogramming assays in the absence of FACT, indicative of higher proliferation rates. In agreement with these reports, our data demonstrate that FACT-depleted ES cells differentiate much more efficiently into early neuronal precursors. Taken together, the data suggest a role for FACT activity during neuronal differentiation, and the proper levels of FACT might assist in balancing proliferation speed and timing of differentiation processes.

# Materials and Methods

### Cell culture

The E14 cell line (mESCs) was cultured at 37°C, 7.5% $CO_2$, on 0.1% gelatin coated plates, in DMEM + GlutaMax (Gibco) with 15% fetal bovine serum (Gibco), MEM nonessential amino acids (Gibco), penicillin/streptomycin (Gibco), 550 μM 2-mercaptoethanol (Gibco), and 10 ng/ml of leukaemia inhibitory factor (eBioscience). HEK293T, N2a, MEFs, NIH3T3, and B16 cell lines were cultured at 37°C, 5% $CO_2$ in DMEM + GlutaMax (Gibco) with 10% fetal bovine serum (Gibco), and penicillin/streptomycin (Gibco). Early NPCs were generated as previously described (Bibel et al, 2007). Briefly, embryoid bodies were created with the hanging drop technique and were further treated with 1 μM RA for 4 d. RA-treated embryoid bodies were trypsinised and cultured in DMEM + GlutaMax (Gibco) with 15% fetal bovine serum without leukaemia inhibitory factor for 3 d.

### Depletion of SSRP1 from mESCs via shRNA and RNA preparation

E14 were transfected with lentiviral vectors containing either a scramble control or *Ssrp1* shRNAs (MISSION shRNA; Sigma-Aldrich) with the following sequences:

**Sequences for E14 transfection.**

| | |
|---|---|
| Scramble control | CCGGGCGCGATAGCGCTAATAATTTCTCGAGAAATTATTAGCGCTATCGCGCTTTTT |
| shRNA 1 (*Ssrp1*) | CCGGCCTACCTTTCTACACCTGCATCTCGAGATGCAGGTGTAGAAAGGTAGGTTTTTG |
| shRNA 2 (*Ssrp1*) | CCGGGCGTACATGCTGTGGCTTAATCTCGAGATTAAGCCACAGCATGTACGCTTTTTG |
| shRNA 3 (*Ssrp1*) | CCGGGCAGAGGAGTTTGACAGCAATCTCGAGATTGCTGTCAAACTCCTCTGCTTTTTG |
| shRNA 4 (*Ssrp1*) | CCGGCCGTCAGGGTATCATCTTTAACTCGAGTTAAAGATGATACCCTGACGGTTTTTG |

A combination of two different *Ssrp1* shRNAs was used (1 and 2; 3 and 4) at a time, and depletion was quantified via western blotting using a monoclonal anti-Ssrp1 antibody (BioLegend). Anti-α tubulin was used as a reference control. The 1 and 2 combination was used for subsequent experiments as it yielded higher depletion of SSRP1 levels (Fig S2A and B). 48 h after transfection, puromycin (2 μg/ml) selection was applied for an additional 24-h period, before cell collection and RNA preparation. Total RNA was obtained via phenol–chloroform extraction (QIAzol Lysis Reagent; QIAGEN) fol-lowed by purification via Quick-RNA MicroPrep (Zymo Research). Library preparation and ribosomal depletion were performed via the NEBNext Directional RNA Ultra kit (NEB) and the RiboZero kit (Illumina), respectively, according to the manufacturer's in-structions. Four different biological replicates (control or SSRP1-depleted mESCs) were prepared and processed for transcriptome analysis.

### MTT proliferation assay

48 h after transfection, different cell densities ($3 × 10^4$, $2 × 10^4$, and $1 × 10^4$) were seeded on 96-well plates (Sarstedt) along with puromycin (2 μg/ml). 24 h later, the CellTiter 96 Non-Radioactive Cell Pro-liferation Assay kit (Promega) was used according to the manufac-turer's instructions to assess the rate of cell proliferation between the two conditions (control and *Ssrp1* KD). Statistical analysis was performed using a two-tailed *t test*.

### Transcriptome analysis in SSRP1-depleted mESCs

Sequenced reads were aligned to the mm10 genome via STAR (v 2.4.1b) (Dobin et al, 2013). Gene and exon counts were obtained from featureCounts of the Rsubread package (R/Bioconductor). Only reads with counts per million > 1 were kept for subsequent analysis. Counts were normalised using the internal TMM nor-malisation in edgeR (Robinson et al, 2009) and differential ex-pression was performed using the limma (Ritchie et al, 2015) package. All of the RNA-seq data presented in this article have been normalised to the total library size. Significant genes with an ab-solute logFC > 1 and adjusted *P*-value < 0.01 were considered as differentially expressed (Table S1). The "unchanged" gene class (*n* = 2,179) was obtained from genes with an adjusted *P*-value > 0.05. The diffSplice function implemented in limma was used to identify differentially spliced exons between the two conditions (Table S2). Significant exons with an FDR < 0.001 were considered as differ-entially spliced. Retention introns were identified using the MISO (Mixture of Isoforms) (Katz et al, 2010) probabilistic framework (Table S3).

### Retention intron events

We verified the presence of retained introns in the *Ssrp1* KD by randomly selecting 10 intron retention events. The FastStart SYBR Green Master (Roche) was used along with the following primers to amplify via PCR the retained intragenic regions:

**Primers for PCR amplification of the retained intragenic regions.**

| Gene name | Forward primer | Reverse primer |
|-----------|----------------|----------------|
| Men1 | ATTTCCCAGCAGGCTTCAGG | GGGATGACACGGTTGACAGC |
| Dvl1 | CCTGGGACTACCTCCAGACA | CCTTCATGATGGATCCAATGTA |
| Map4k2 | GCTGCAGTCAGTCCAGGAGG | TCCTGTTGCTTCAGAGTAGCC |
| Ctsa | GCAATACTCCGGCTACCTCA | TGGGGACTCGATATACAGCA |
| Pol2ri | CGAAATCGGGAGTGAGTAGC | GGTGGAAGAAGGAACGATCA |
| Wipf2 | TAGAGATGAGCAGCGGAATC | TCGAGAGCTGGGGACTTGCA |
| Fuz | GACCCAGTGTGTGGACTGTG | GACAAAGGCTGTGCCAGTGG |
| Rfx5 | CACCAGTTGCCCTCTCTGAA | CAATTCTCTTCCTCCCATGC |
| Fhod1 | CACCAGGGAGCAGAGATGAT | CCATCAACATTGGCCTAACC |
| Tcirg1 | AGCGACAGCACTCACTCCTT | CAACACCCCTGCTTCCAGGC |

Amplified products were run on a 1.5% agarose gel and visualised under UV. Band quantification was performed with ImageJ.

### ChIP of FACT subunits

ChIP was performed in ~20 million ES cells, per assay, as described previously (Tessarz et al, 2014) with a few modifications. Briefly, cells were cross-linked with 1% formaldehyde for 20 min followed by quenching for 5 min with the addition of glycine to a final concentration of 0.125 M. After washing with PBS buffer, the cells were collected and lysed in cell lysis buffer (5 mM Tris, pH 8.0, 85 mM KCl, and 0.5% NP-40) with proteinase inhibitors (10 µl/ml phenyl-methylsulfonyl fluoride, 1 µl/ml leupeptin, and 1 µl/ml pepstatin). Pellets were spun for 5 min at 5,000 rpm at 4°C. Nuclei were lysed in nuclei lysis buffer (1% SDS, 10 mM EDTA, and 50 mM Tris–HCl) and samples were sonicated for 12 min. The samples were centrifuged for 20 min at 13,000 rpm at 4°C and the supernatant was diluted in IP buffer (0.01% SDS, 1.1% Triton-X-100, 1.2 mM EDTA, 16.7 mM Tris–HCl, and 167 mM NaCl), and the appropriate antibody was added and left overnight with rotation at 4°C. Anti-Ssrp1 and anti-Supt16 antibodies were purchased from BioLegend (#609702) and Cell Signalling (#12191), respectively. Anti-AP-2γ (Tfap2c) antibody was purchased from Santa Cruz (#sc-12762). Two biological replicates were prepared for each FACT subunit using independent cell cultures and chromatin precipitations. Protein A/G Dynabeads (Invitrogen) were added for 1 h and after extensive washes, the samples were eluted in elution buffer (1% SDS and 0.1 M NaHCO$_3$). 20 µl of 5 M NaCl was added and the samples were reverse cross-linked at 65°C for 4 h. Following phenol–chloroform extraction and ethanol precipitation, DNA was incubated at 37°C for 4 h with RNAse (Sigma-Aldrich).

### ChIP-seq library preparation, sequencing, and peak calling

Approximately 10–20 ng of ChIP material was used for library preparation. End repair and adaptor ligation was prepared as described previously with a few modifications (Tessarz et al, 2014). Double-sided size selections (~200–650 bp) were performed using the MagSI-NGS Dynabeads (#MD61021; MagnaMedics) according to the manufacturer's instructions. Purified adapter-ligated

ChIP material was run on a high-sensitivity DNA chip on a 2200 TapeStation (Agilent) to assess size distribution and adaptor contamination.

The samples were single-end deep-sequenced and reads were aligned to the mm10 genome using Bowtie2 (v 2.2.6) (Langmead & Salzberg, 2012). Peak calling was performed using PePr (v 1.1) (Zhang et al, 2014) with peaks displaying an FDR < 10$^{-5}$ considered statistically significant (Table S4). Peak annotation was performed via the ChIP-Enrich (Welch et al, 2014) R package with the following parameters (locusdef = "nearest_gene" and method = "broadenrich").

### ChIP-seq normalisation and metagene analysis

All the ChIP-seq BAM files were converted to bigwig (10 bp bin) and normalised to 1× sequencing depth using deepTools (v 2.4) (Ramirez et al, 2016). Blacklisted mm9 coordinates were converted to mm10 using the LiftOver tool from UCSC and were further removed from the analysis. Average binding profiles were visualised using R (v 3.3.0). Heatmaps were generated via deepTools. For the average profiles in Fig S1C and D, RPKM values from control ES RNA-seq data were divided into four different quantiles and the average profile for each FACT subunit was generated for each quantile. The Pearson's correlation plot in Fig 1A was generated using all unique annotated mm10 RefSeq genes (n = 11,305) from UCSC (blacklisted regions were removed).

### MNase-seq following SSRP1 depletion in mESCs

ES cells were cultured and transfected with shRNA vectors as described above. Biological replicates were obtained from two independent transfection experiments for each shRNA vector. Briefly, ~5 million cells were cross-linked with 1% formaldehyde for 20 min followed by quenching for 5 min with the addition of glycine to a final concentration of 0.125 M. After washing with PBS buffer, the cells were collected and lysed in cell lysis buffer (5 mM Tris, pH 8.0, 85 mM KCl, and 0.5% NP-40) with proteinase inhibitors (10 µl/ml phenylmethylsulfonyl fluoride, 1 µl/ml leupeptin, and 1 µl/ml pepstatin). Nuclei were gathered by centrifugation (5,000 rpm for 2 min) and were treated with 10 Kunitz units/10$^6$ cells of micro-coccal nuclease (#M0247S; NEB) for 5 min at 37°C in 40 µl of micrococcal nuclease buffer (#M0247S; NEB). The reaction was stopped with the addition of 60 µl 50 mM EDTA, 25 µl 5 M NaCl, and 15 µl 20% NP-40 and incubated on a rotator for 1 h at room temperature to release soluble nucleosomes. The samples were centrifuged for 5 min at 10,000 g and the supernatant was transferred to a new tube. This centrifugation step is important to obtain highly soluble nucleosomes and remove nucleosome–protein complexes, which can raise bias in subsequent data interpretation (Carone et al, 2014) (Fig S7). The samples were reverse cross-linked by incubating overnight at 65°C with 0.5% SDS and proteinase K. Following phenol–chloroform extraction and ethanol precipitation, DNA was incubated at 37°C for 4 h with RNAse (Sigma-Aldrich). All samples were run in a 2% agarose gel and fragments <200 bp were extracted and purified using the NucleoSpin Gel and PCR Clean-up kit (Macherey-Nagel) according to the manufacturer's instructions.

 **Life Science Alliance**

Purified DNA (500 ng) was used for library preparation as described above. The only difference was the PCR amplification step where we used the same conditions as mentioned in Henikoff et al (2011) but with only three amplification cycles. Libraries were verified using a 2200 TapeStation and were paired-end deep-sequenced (~250 million reads per sample). For quality checks and reproducibility, please refer to Fig S7.

## MNase-seq normalisation and metagene analysis

All the MNase-seq BAM files were converted to bigwig, binned (1 bp), smoothed (20-bp window), and normalised to 1× sequencing depth using deepTools (v 2.4). Moreover, they were split into two different categories according to fragment length: <80 bp TF-sized fragments and 135–170 bp mononucleosome fragments. Average nucleosome occupancy profiles were visualised using R (v 3.3.0). For the Fig S7D and E, the mm10 annotated exon list for mononucleosomal profiling was obtained from UCSC.

## ATAC-seq following SSRP1 depletion in mESCs

ES cells were cultured and transfected with shRNA vectors as described above. Biological replicates were obtained from two independent transfection experiments for each shRNA vector. ATAC-seq was performed on 50,000 cells as previously described (Buenrostro et al, 2013). All samples were PCR amplified for nine cycles and were paired-end sequenced on an Illumina HiSeq 2500 platform.

## ATAC-seq normalisation and metagene analysis

Sequenced paired mates were mapped on mm10 genome build using Bowtie2 with the following parameters: –X 2000. Reads corresponding to NFRs were selected via a random forest approach using the "ATACseqQC" R package. All the ATAC-seq BAM files were converted to bigwig, binned (1 bp), and normalised to 1× sequencing depth using deepTools (v 2.4). Duplicated reads were removed. Chromatin accessibility profiles were visualised using R (v 3.3.0).

## Mass spectrometry sample preparation and analysis

Nuclei were isolated from ~5 million ES cells under hypotonic conditions and the samples were incubated overnight at 4°C with an anti-H3K4me3 antibody (#39159; Active Motif) in the presence of low-salt binding buffer (150 mM NaCl, 50 mM Tris–HCl, pH 8.0, and 1% NP-40), protease inhibitors, and Protein G Dynabeads (Invitrogen). The following day, after several rounds of bead washing with binding buffer, the samples were incubated overnight at 37°C with Tris, pH 8.8, and 300 ng Trypsin Gold (Promega). In total, four samples were prepared for each condition (control and *Ssrp1* KD). For the full protein interactome of both FACT subunits, nuclei were extracted as described above, and anti-Ssrp1 and anti-Supt16 antibodies were used. Peptides were desalted using StageTips (Rappsilber et al, 2003) and dried. The peptides were resuspended in 0.1% formic acid and analyzed using liquid chromatography—mass spectrometry (LC-MS/MS).

## LC-MS/MS analysis

For mass spectrometric analysis, the peptides were separated online on a 25-cm 75 µm ID PicoFrit analytical column (New Objective) packed with 1.9 µm ReproSil-Pur media (Dr. Maisch) using an EASY-nLC 1000 (Thermo Fisher Scientific). The column was maintained at 50°C. Buffer A and B were 0.1% formic acid in water and 0.1% formic acid in acetonitrile, respectively. The peptides were separated on a segmented gradient from 5% to 25% buffer B for 45 min, from 25% to 35% buffer B for 8 min, and from 35% to 45% buffer B for 4 min at 200 nl/min. Eluting peptides were analyzed on a QExactive HF mass spectrometer (Thermo Fisher Scientific). Peptide precursor mass to charge ratio ($m/z$) measurements (MS1) were carried out at 60,000 resolution in the 300 to 1,500 $m/z$ range. The top 10 most intense precursors with charge state from two to seven only were selected for HCD fragmentation using 27% collision energy. The $m/z$ of the peptide fragments (MS2) were measured at 15,000 resolution, using an AGC target of 1e6 and 80 ms maximum injection time. Upon fragmentation, the precursors were put on an exclusion list for 45 s.

## LC-MS/MS data analysis

The raw data were analysed with MaxQuant (Cox & Mann, 2008) (v 1.5.2.8) using the integrated Andromeda search engine (Cox et al, 2011). Fragmentation spectra were searched against the canonical and isoform sequences of the mouse reference proteome (proteome ID: UP000000589, downloaded in August 2015) from UniProt. The database was automatically complemented with sequences of contaminating proteins by MaxQuant. For the data analysis, methionine oxidation and protein N-terminal acetylation were set as variable modifications. The digestion parameters were set to "specific" and "Trypsin/P," allowing for cleavage after lysine and arginine, also when followed by proline. The minimum number of peptides and razor peptides for protein identification was 1; the minimum number of unique peptides was 0. Protein identification was performed at a peptide spectrum match and protein false discovery rate of 0.01. The "second peptide" option was on to identify co-fragmented peptides. Successful identifications were transferred between the different raw files using the "match between runs" option, using a match time window of 0.7 min. Label-free quantification (LFQ) (Cox et al, 2014) was performed using an LFQ minimum ratio count of 2.

## Identification of co-enriched proteins

Analysis of the LFQ results was carried out using the Perseus computation platform (Tyanova et al, 2016) (v 1.5.0.0) and R. For the analysis, LFQ intensity values were loaded in Perseus and all identified proteins marked as "Reverse," "Only identified by site," and "Potential contaminant" were removed. Upon log2 transformation of the LFQ intensity values, all proteins that contained less than four missing values in one of the groups (control or *Ssrp1* KD) were removed. Missing values in the resulting subset of proteins were imputed with a width of 0.3 and down shift of 1.8. Next, the imputed LFQ intensities were loaded into R where a two-side testing for enrichment was performed using limma (Kammers et al, 2015; Ritchie et al, 2015). Proteins with an adjusted *P*-value < 0.05 were designated as significantly enriched in the control or knockdown (H3K4me3 IP)

(Table S5). The complete list of differential protein expression between control and *Ssrp1* KD can be found in Table S6.

## NET-seq library preparation

ES cells were cultured and transfected with shRNA vectors as described above. Biological replicates were obtained from two independent transfection experiments for each shRNA vector. NET-seq libraries were prepared as previously described (Mayer & Churchman, 2016) with a few modifications. Briefly, chromatin-associated nascent RNA was extracted through cell fractionation in the presence of α-amanitin, protease, and RNAase inhibitors. More than 90% recovery of ligated RNA and cDNA was achieved from 15% TBE-Urea (Invitrogen) and 10% TBE-Urea (Invitrogen), respectively, by adding RNA recovery buffer (R1070-1-10; Zymo Research) to the excised gel slices and further incubating at 70°C (1,500 rpm) for 15 min. Gel slurry was transferred through a Zymo-Spin IV Column (C1007-50; Zymo Research) and further precipitated for subsequent library preparation steps. cDNA containing the 3′ end sequences of a subset of mature and heavily sequenced snRNAs, snoRNAs, and rRNAs were specifically depleted using biotinylated DNA oligos (Table S7). Oligo-depleted circularised cDNA was amplified by PCR (five cycles) and double-stranded DNA was run on a 4% low melt agarose gel. The final NET-seq library running at ~150 bp was extracted and further purified using the ZymoClean Gel DNA recovery kit (Zymo Research). Sample purity and concentration was assessed in a 2200 TapeStation and further deep sequenced in a HiSeq 2500 Illumina Platform (~400 million reads per replicate).

## NET-seq analysis

All the NET-seq FASTQ files were processed using custom Python scripts (https://github.com/BradnerLab/netseq) to remove PCR duplicates and reads arising from RT bias. Reads mapping exactly to the last nucleotide of each intron and exon (splicing intermediates) were further removed from the analysis. The final NET-seq BAM files were converted to bigwig (1 bp bin), separated by strand, and normalized to 1× sequencing depth using deepTools (v 2.4) with an "–offset 1" to record the position of the 5′ end of the sequencing read. NET-seq tags sharing the same or opposite orientation with the TSS were assigned as "sense" and "antisense" tags, respectively. Promoter-proximal regions were carefully selected for analysis to ensure that there is minimal contamination from transcription arising from other transcription units. Genes overlapping within a region of 2.5 kb upstream of the TSS were removed from the analysis. For the NET-seq metaplots, genes underwent several rounds of k-means clustering to filter regions; in a 2-kb window around the TSS, rows displaying very high Pol II occupancy within a <100-bp region were removed from the analysis as they represent non-annotated short noncoding RNAs. Average Pol II occupancy profiles were visualised using R (v 3.3.0). In Fig 4B, the proximal promoter region was defined as −30 bp and +250 bp around the TSS. For Fig 4A and B, gene body coverage was retrieved by averaging all regions (FACT-bound and non–FACT-bound) +300 bp downstream of TSS and −200 bp upstream of transcription end site. Comparison

of the two linear regressions was performed by calculating the z-score by the following equation:

$$z = \frac{\beta_1 - \beta_2}{\sqrt{s_{\beta_1}^2 + s_{\beta_2}^2}}$$

where $\beta$ and $s_\beta$ represent the "slope" and the "standard error of the slope," respectively. *P*-value was calculated from the respective confidence level yielded by the z-score.

## Immunofluorescence and confocal microscopy

Early NPCs were generated and *Ssrp1* levels were knocked down as described above. The cells were fixed with 100% ethanol for 10 min and processed for immunofluorescence. Permeabilization and blocking was performed for 1 h at room temperature with 1% BSA and 0.1% NP-40 in PBS. Incubation with primary antibodies was carried at room temperature for 2 h by using rabbit anti-β3-tubulin (1:300; Cell Signaling) and mouse anti-MAP2 (1:300; Millipore). After washing in blocking buffer, the secondary antibodies anti-rabbit and anti-mouse Alexa Fluor 568 (1:1,000; Life Technologies) were applied for 2 h at room temperature. Slides were extensively washed in PBS and nuclei were counterstained with DAPI before mounting. Fluorescence images were acquired using a laser scanning confocal microscope (TCS SP5-X; Leica), equipped with a white light laser, a 405-diode UV laser, and a 40× objective lens.

## Gene ontology analysis

All GO terms were retrieved from the metascape online platform (http://metascape.org/).

## Accession numbers and references of publicly available datasets

H3K4me3, H3K27me3, Pol II S5ph, H3K4me1, H3K27Ac, and CTCF (ENCODE Consortium—E14 cell line); Chd1 and Chd2 (de Dieuleveult et al, 2016): GSE64825; p53 (Li et al, 2012): GSE26360; and Pol II S2ph (Brookes et al, 2016): GSM850470. Data generated in this study have been deposited in Gene Expression Omnibus (GEO) under accession number GSE90906 (ChIP-seq, RNA-seq, chrRNA-seq, MNase-seq, ATAC-seq, and NET-seq).

# Supplementary Information

# Acknowledgements

We would like to thank Ilian Attanassov of the Max Planck Institute for Biology of Ageing (MPI-AGE) Proteomics Core Facility for Mass Spectrometry Analysis and the FACS and Imaging Facility for help with microscopy. We are particularly grateful to Franziska Metge and Sven Templer (MPI-AGE Bioinformatics Core) for their assistance with coding script formatting. Sequencing was performed at the Max Planck Genome core centers in Berlin

and Cologne, and data analysis was performed on servers of the GWDG, Göttingen, and the MPI-AGE cluster. We thank Andy Bannister, Antonis Kirmizis, and members of the Tessarz laboratory for discussion and comments on the manuscript. This work was funded by the Max Planck Society.

## Author Contributions

P Tessarz: conceptualization, supervision, funding acquisition, project administration, and writing—review and editing.
C Mylonas: resources, data curation, software, formal analysis, validation, investigation, visualization, and writing—original draft, review, and editing.

## Conflict of Interest Statement

The authors declare that they have no conflict of interest.

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
