## [Reviewer comments · Life Science Alliance]

Transcriptional repression by FACT is linked to regulation of chromatin accessibility at the promoter of ES cells

Constantine Mylonas and Peter Tessarz

DOI: 10.26508/lsa.201800085

Review timeline:	Submission date:	8 May 2018
	1 st Revision received:	8 May 2018
	1 st Editorial Decision:	16 May 2018
	2 nd Revision received:	28 May 2018
	2 nd Editorial Decision:	28 May 2018
	3 rd Revision received:	29 May 2018
	Accepted:	29 May 2018

Report:

(Note: Letters and reports are not edited. The original formatting of letters and referee reports may not be reflected in this compilation.)

1st Revision – authors' response

8 May 2018

Referee #1: Mylonas et al. investigate the role of FACT in mESCs. The authors profiled the genomic localization of its two complex members, SSRP1 and SUPT16, and assayed the transcriptional consequences upon depletion of SSRP1. MNase-, ATAC- and NET-Seq in SSRP1-depleted cells revealed a loss of nucleosomal occupancy and gain of chromatin accessibility as well as concomitant increase in antisense transcription at TSSs of upregulated SSRP1-target genes. Although, the authors cannot "determine if the loss of nucleosomes precedes upregulation of antisense transcription or vice-versa" and whether the change is "driven by FACT alone, or in combination with RNA polymerase and/or chromatin remodellers", they suggest that "FACT might act as a repressor by enabling a more closed chromatin conformation state at promoter regions".

At this point, we are unable to recommend publication of this manuscript in EMBO J, because:

1. The authors do not provide a major understanding for how FACT regulates transcription. Is it through nucleosome occupancy? If so, how? Is it through regulating transcription? If so, how? In other words, the authors have provided suggestions for how FACT control transcription, which are based on relatively few results.
2. Does FACT have a role in regulating cell fate decision? Most likely not, but it may have a role in maintaining the proper gene expression program in ES cells (and other cell types), and downregulation of FACT together with a directed differentiation program may therefore lead to an increased number of differentiated cells. In any case, the observation that downregulation of FACT leads to an increase in neural differentiation is not supported by data demonstrating a particular role of FACT in this process.
3. The authors write that "FACT can work directly as an enhancer or repressor of transcription in mES cells." They do not provide results or suggestions for how FACT should work in enhancing transcription.

In addition to the lack of major insight into FACT function, there are a number of additional concerns regarding the manuscript:

1. The experiments lack essential controls:

a) ChIPseq: To verify the identified target loci with anti-SSRP1 and anti-SUPT16 antibodies, KD or KO controls need to be used. Input (and IgG) controls from the same chromatin preparation should also be sequenced. Furthermore, the authors should use a more quantitative assay like ChIP-qPCR on a number of the identified targets to validate the ChIPseq results.

b) MTT assay/RNAseq: To account for potential shRNA-specific off-targets, rescue experiments should be performed using shRNA-resistant SSRP1 expressing cells.

2. The authors claim that they are investigating the function of FACT throughout the manuscript while for most of the studies, this is assessed only through loss of SSRP1. As mentioned in the abstract, FACT is a heterodimer, and therefore loss of SUPT16 should phenocopy the conclusions drawn by studying SSRP1. It should be stated how many target sites of SSRP1 and SUPT16 overlap and how many of the up- and downregulated regulated genes upon SSRP1-depletion are direct targets of FACT. Only target sites that are co-occupied by SUPT16 and SSRP1 should be considered as "FACT-bound". Also, the data in Fig 2 would be supported by assessing the phenotypic and transcriptional consequences through downregulation of SUPT16 and overlapping with those observed by SSRP1 KD.

Minor points:

1. SSRP1 mRNA expression levels should be corroborated by protein expression levels.

2. Fig 1C: The authors should show representative tracks of the ChIP data.

3. The authors state a "hyper-proliferative" phenotype in mESCs upon FACT-depletion. However, this is only assessed through MTT assays, which indirectly measure proliferation by assessing metabolic activity. Therefore, the data should be corroborated by actual growth curves.

4. Fig 2D: Instead of the depicted figure, it would be more descriptive to state the number of up- and downregulated genes, respectively, that are direct targets identified by ChIPSeq.

5. It is not clear how many differentially spliced genes are direct targets of FACT, which is needed to support the hypothesis that FACT recruits splice factors to H3K4me3 positive regions.

6. The authors speculate that "up-regulated genes showed a loss of nucleosome occupancy in the gene body area regardless of FACT-bound status ..., potentially reflecting the higher transcription rate through these genes". However, in order to make this point, all genes should be ranked by their transcription rate and blotted against their nucleosome occupancy. If the transcriptional rate correlates with nucleosome occupancy independently of FACT, differences should be observable even within control cells (not only between control and SSRP1 KD cells).

7. In order to confirm the observation in Fig 3A (SSRP1-target specific loss of nucleosomes at TSS of up-regulated genes) it is essential to show ATAC-Seq profiles of down- and up-regulated genes of non-SSRP1 targets in Fig 3C.

8. Fig 3D is difficult to interpret and doesn't give additional information to Fig 3C and should therefore be removed.

9. Besides the stated points, it seems counterintuitive to investigate the differentiation of ES cells into the neuronal lineage upon FACT-depletion since mRNA levels of both FACT complex members don't change during differentiation (Fig 5A). It would seem more interesting to investigate the contribution of FACT in e.g. cardiomyocyte differentiation for which changes in FACT are observed. Focusing on neuronal differentiation, the authors could investigate FACT occupancy in neural progenitor cells at genes that are targeted by FACT and upregulated by its loss in mESCs to strengthen the point that FACT primes ES cells for neuronal lineage differentiation.

Revisions based on Reviewer 1:

- we have now added a ChIP-seq example as part of new Figure 1D
 - we also added the numbers of up- and downregulated genes into new Figure 2D
 - we have added a half sentence into the manuscript with respect to the direct targets of misspliced transcripts: 'of which around 50% are direct targets of SSRP1'
 - we removed the sentence: 'potentially reflecting the higher transcription rate through these genes' as this was too speculative
-

Point-by-point answers to Reviewer 2:

The authors examined the role of the murine FACT complex in development with an emphasis on embryonic stem cells. Reduction of FACT is associated with differentiation and can be seen in the transition from mouse embryonic stem cells (mESCs) to cardiomyocytes. They start by using ChIP-seq to identify where the two subunits of FACT (SSRP1 and SUPT16) bind to genomic DNA and RNA-seq to confirm their effects on gene expression. Next changes in nucleosome occupancy and DNA accessibility when FACT is depleted are examined using MNase-seq and ATAC-seq. The authors further check whether FACT is involved in controlling RNA polymerase II promoter proximal pausing with NET-seq; which maps both paused, stalled and elongating RNA polymerase II. The last part of the paper is loosely connected to the prior data and examines the effects when mESCs are differentiated into neuronal progenitor cells (NPC) and FACT is depleted. It is important to establish if there is a connection between FACT and transcription pausing and directionality particularly in pluripotency, a main thrust of this paper, and would be a critical finding.

The idea behind the differentiation assay was to test if FACT is required to maintain an ES cells transcriptome. AS E14 ES cells (used in this study) are prone to differentiate into the neuronal lineage, we chose to test this. This decision was also based on the isoform analysis (Supplementary Figure 3) and previous hints in the literature (as discussed in the Discussion section) that involved FACT in the process of neurogenesis.

FACT appears to be localized near the transcription start sites (TSSs) and to be correlated with marks of active transcription such as H3K4me3, H3K27ac and phosphorylated forms of RNAP II (Ser5). Previously CHD1 and FACT have been shown to physically interact and the authors suggest this is consistent with the ChIP-seq data; however, their binding sites are actually shown to be mutually exclusive, while still being proximal. I wasn't sure how the authors can explain this mutual exclusivity phenomena. Although the ChIP-seq suggest an activation role for FACT in transcription, gene expression shows that with FACT depletion genes are almost as likely to be activated or repressed and suggest a more complex relationship of FACT to transcription. On page 7 lines 155-159, the authors seem to indicate that transcription factors Oct4 and Sox2 plus splicing factors directly recognize and bind to H3K4me3. Is that really what the authors intended and if so what is the evidence for direct binding versus merely a correlation between where they bind in the genome? The changes in DNA accessibility observed when SSRP1 is depleted are not very large even though they are statistically significant in the case of those genes that are up-regulated and are non-existent for those genes down-regulated. I was concerned about the lack of reproducibility of the MNase-seq data as seen in Supplementary Figure 4C. In the two replicates for Ssrp1 KD (two different shades of red) one of the replicates looks essentially the same as the control and the other shows the desired difference. This data does not look at all very convincing and am deeply concerned about reproducibility.

We do not understand this comment. It refers to Supplementary Figure 4C and to us it is very clear that while there is variability in the gene body area, the data for the TSS is very reproducible. This is one of the reasons why we focused on the TSS (where FACT binding also peaks in mammals). We would also like to highlight that the described class of genes (up-regulated and FACT-bound) are a very diverse class of genes and comprise of genes that are usually silenced to highly expressed genes. This is indicated by the three different classes in Supplementary Figure 4C. It is very obvious that genes that used to be more repressed also show higher changes in the nucleosome occupancy. We have analysed all of our data by splitting them into these three classes (and are happy to provide them), but felt that it makes the manuscript more complicated to follow.

In addition, we provide quality control data for the MNase-seq experiment based on previously published datasets (Supplementary Figure 7).

This is the only case in which replicates are actually provided and highlights a critical deficiency in the paper.

Again, we feel that this is an unjustified comment as we provide correlation plots for all sequencing replicates that demonstrate very high correlations (ATAC-seq: Supplementary Figure 4C – R=0.989 for CTRL and R=0.988 for the knock down; NET-seq: Supplementary Figure 5A – R=0.97 for CTRL and R=0.99 for knock down). Particularly in the case of NET-seq, replicates are usually pooled to provide the necessary coverage.

In lines 186-187 it states there is a difference in between up- and down-regulated genes, but in Supplementary Figure 4C, class 2 did not show a significant difference between these two classes of genes.

In this Figure, we do not compare up- and down-regulated genes, but simply compare the nucleosome profile at the TSS of genes in the presence and absence of FACT. ALL of these genes are up-regulated in the absence of SSRP1.

While the difference in ATAC signal seen in Figure 3C for up-regulated is statistically significant, it is not clear as to its biological significance given the small magnitude of the change. I think the conclusion based on this data of FACT acting as a repressor to promote a more closed chromatin structure is tenuous.

Similar to the argument in the case for MNase-seq, this is a metaplot of all up-regulated genes comprising of very different ground states. Another complication with ATAC-seq is the fact that it also takes into account transcription factor binding. We have split now the data into smaller and larger fragments and used an algorithm developed by the Greenleaf laboratory to calculate nucleosomal density (Reviewer Figure 1). This nicely supports the finding of the MNase-seq. However, we would prefer not to include this data into the manuscript.

The data in this paper suggests that FACT does not control the release of RNAPII pausing as seen by measuring the travelling ratio or pause index with and without depletion of SSRP. However, based on density plots of the NET-seq data the authors conclude that there is more RNAPII pausing in the sense direction of SSRP targets when SSRP1 is depleted (see Figure 4C). I am not sure if there is a clear basis only in the sense direction as it appears there is also an increase in the anti-sense direction of SSRP targets when SSRP1 is depleted and although the overall levels are different between sense and anti-sense, the relative change in the anti-sense with and without SSRP1 are as significant as for the sense transcripts. The authors need to comment more on how these data are normalized as this appears to be particularly difficult for NET-seq type data and is a critical issue for the type of comparisons the authors are making. The single example provided of Oct4 as an SSRP target is not convincing because the changes in NET-seq signal between the control and Ssrp1 KD is not significantly different. I did not find the data in the paper very compelling in regards to the conclusions that FACT is linked to nucleosome deposition at the promoter and the obstruction of anti-sense transcription.

We thank the reviewer for this observation. Indeed, there is an increase in RNA Pol II pausing in both direction. We have deleted the sentence: 'in the sense strand, but not in the antisense strand'. We have described in the Methods how the NET-seq data was normalized and averaged: 'For the NET-seq metaplots, genes underwent several rounds of k-means clustering in order to filter regions; in a 2kb window around the TSS, rows displaying very high Pol II occupancy within a <100 bp region were removed from the analysis as they represent non-annotated short non-coding RNAs. Average Pol II occupancy profiles were visualised using R (v 3.3.0). In Figure 4B, the Proximal Promoter region was defined as -30 bp and +250 bp around the TSS. For Figure 4A,B, gene body coverage was retrieved by averaging all regions (FACT-bound and non-FACT-bound) +300 bp downstream of TSS (Transcription Start site) and -200 bp upstream of TES (Transcription End Site).' We also toned down our statement about the role of SSRP1 in promoter release and simply described our data: 'Indeed, "up-regulated" SSRP1-bound genes show a lower travelling ratio overall. Interestingly, under our experimental conditions, we did not observe a significant difference among this group of genes following FACT depletion (Control to *Ssrp1* KD comparison – Fig. 4B).'

There are several minor points that also need to be mentioned. I did not find any reference in the text to the data shown in Supplementary Figure 7, which needs to be corrected. In Figure 2D the authors may want to explain the meaning of the gray area and the dashed line. The Supplementary Figure 3 has a total of five panels in the legend but only four panels in the figure. The panel B appears to be missing based on the figure legend.

All fixed.

1st Editorial Decision

16 May 2018

Thank you for submitting your manuscript entitled "Transcriptional repression by FACT is linked to regulation of chromatin accessibility at the promoter of ES cells" to Life Science Alliance. Your manuscript was reviewed at another journal before, and the editors have transferred the previous referee reports to us with your permission. You submitted to us a revised manuscript and a point-by-point response to the concerns raised during this previous round of peer-review.

I wanted to involve one of the original reviewers in the re-review of your work, but unfortunately, the reviewer could not help this time. I therefore decided to seek arbitrating advice on your revised manuscript. I have now heard back from the advisor.

The advisor appreciates the diverse datasets and the good reproducibility provided. However, the advisor thinks that a few issues that were previously raised would need your further attention before moving towards publication in Life Science Alliance:

- the lack of in-depth understanding of mechanisms at play needs to be clearly mentioned, i.e. the correlations provided need to be highlighted as such (as an example, please state that it is unclear at this stage whether the change of splicing pattern and the change of splicing factor bound with H3K4me3 after perturbing FACT are mechanistically coupled).
- the part on FACT depletion during neuronal differentiation will need some text changes as well. The readers should get informed about FACT levels not changing during differentiation, please rewrite the part accordingly.
- the major effects of FACT on ES proliferation being due to regulation at TSS should get further toned-down (also in the model figure), as a large number of gene expression changes could not be explained by this.
- finally, the advisor also thinks that genomics approaches are prone to random noise, but also systematic ones, and that a comparison between Mnase-Seq and ATAC-seq data would have further ensured the quality of your dataset. This point was not raised in the initial round of peer-review, and addressing this point is therefore not mandatory for publication here. But it seems that the advisor raised a constructive point, and I therefore decided to give you the option to address it.

I would thus like to ask you to provide a final revision of your manuscript, addressing the first three points mentioned above and to consider addressing the last point.

Thank you for this interesting contribution to Life Science Alliance. We are looking forward to receiving your revised manuscript.

2nd Revision – authors' response

28 May 2018

Point-by-point answers to the referee's comments (in bold):

- the lack of in-depth understanding of mechanisms at play needs to be clearly mentioned, i.e. the correlations provided need to be highlighted as such (as an example, please state that it is unclear at this stage whether the change of splicing pattern and the change of splicing factor bound with H3K4me3 after perturbing FACT are mechanistically coupled).

For the splicing section, we added a sentence outlining that the present dataset only allows correlations to be made (line 159): *However, at present it is not clear, whether the effects on splicing factor binding and splicing pattern are directly and mechanistically coupled to FACT depletion.*

We changed the sub-heading for one of the parts to highlight that the observed increase in antisense transcription and increased chromatin accessibility is only correlated (line 199/200): *Gain in chromatin accessibility upon FACT depletion upstream of the TSS correlates with an increase in antisense transcription*

We deleted the following sentence at the end of this paragraph (line 237): *Taken together, our data suggests that the repressive function of FACT is linked to nucleosome deposition at the promoter and obstruction of antisense Transcription*

We deleted a similar comment in the Discussion section (line 293: *in which the histone chaperone operates as a repressor, suggesting that FACT is required to maintain the observed high level of nucleosome occupancy and to inhibit antisense transcription.*) and now state: *However, it is clear that this observed effect is very specific to SSRP1-bound genes that are upregulated upon depletion of FACT.*

- the part on FACT depletion during neuronal differentiation will need some text changes as well. The readers should get informed about FACT levels not changing during differentiation, please re-write the part accordingly.

This was already part of the text, but we changed the text now to make it clearer (251-254): *We identified that in these early stage NPCs, expression of key neurogenesis markers (Pax6, Nes, Tubb3) increases whereas FACT mRNA levels and key pluripotency factors are yet unchanged and still maintained at a high level (Fig. 5A).*

- the major effects of FACT on ES proliferation being due to regulation at TSS should get further toned-down (also in the model figure), as a large number of gene expression changes could not be explained by this.

Here, we deleted the sentence and the corresponding Figure 6 that highlighted this link to the TSS: *In ES cells, genes repressed by FACT in this way encode for proteins involved in embryogenesis, particularly in early neuronal differentiation, which is accelerated when FACT is depleted (Fig. 6).*

- finally, the advisor also thinks that genomics approaches are prone to random noise, but also systematic ones, and that a comparison between Mnase-Seq and ATAC-seq data would have further ensured the quality of your dataset. This point was not raised in the initial round of peer-review, and addressing this point is therefore not mandatory for publication here. But it seems that the advisor raised a constructive point, and I therefore decided to give you the option to address it.

As mentioned already in the cover letter, we think that this is beyond the scope of this manuscript and would warrant a systematic study on its own.

2nd Editorial Decision

28 May 2018

Thank you for submitting your revised manuscript entitled "Transcriptional repression by FACT is linked to chromatin accessibility at the promoter of ES cells". I appreciate the introduced changes and am happy to accept your manuscript in principle for publication in Life Science Alliance. Congratulations on this very nice work!

3rd Editorial Decision

29 May 2018

Thank you for submitting your Research Article entitled "Transcriptional repression by FACT is linked to chromatin accessibility at the promoter of ES cells". It is a pleasure to let you know that your manuscript is now accepted for publication in Life Science Alliance. Congratulations on this interesting work.